# Multilingual Abstractive Event Extraction for the Real World

## Abstract

Event extraction (EE) is a valuable tool for making sense of large amounts of unstructured data, with a wide range of real-world applications, from studying disease outbreaks to monitoring political violence. Current EE systems rely on cumbersome mention-level annotations, and event arguments are frequently restricted to ungrounded spans of text, which hinders the aggregation and analysis of extracted events. In this paper, we define a new *abstractive* event extraction (AEE) task that moves away from the surface form and instead requires a deeper wholistic understanding of the input text. To support research in this direction, we release a new multilingual, expert-annotated event dataset called LEMONADE, which covers 16 languages, including several for which no event dataset currently exists. LEMONADE has $41,148$ events, and is based on the Armed Conflict Location and Event Data Project, which has been collecting and coding data on political violence around the globe for over a decade. We introduce a novel zero-shot AEE system ZEST that achieves a score of $57.2\%$ $F_1$ on LEMONADE. With our supervised model that achieves $71.6\%$ $F_1$, they represent strong baselines for this new dataset.

## 1 Introduction

Event extraction (EE) is an important tool for studying the real world. Its applications span a wide range of fields, from social sciences (Zubiaga et al., 2014) to biomedicine (Lybarger et al., 2021; Kim et al., 2003). It is used for early detection and tracking of disease outbreaks (Parekh et al., 2024; Consoli et al., 2024; Min et al., 2021), monitoring cybersecurity threats (Satyapanich et al., 2020), studying political conflicts (Hu et al., 2022), protests (Radford, 2020; Zhukov et al., 2019; Hürriyetoğlu et al., 2022a; Zavarella et al., 2022), and crime (Mostafazadeh Davani et al., 2019). Because of its costly annotation process, automated EE systems are highly desirable.

In AI research, automated event extraction has been an extensively studied topic in information extraction (Ji & Grishman, 2008). However, the resulting EE systems have several shortcomings that keep them from real-world applications (Hürriyetoğlu, 2021; Hürriyetoğlu et al., 2022b; 2023; 2024a). Monitoring socio-political developments perhaps best exemplifies the requirements of event extraction (automated or not) in the real world.

**Entity Normalization and Linking** One of the main uses of event data is trend discovery and aggregate reporting (Li et al., 2019a; 2020b; 2021a; Reddy et al., 2023). Traditional EE systems, which construct extractions based on text spans (Huang et al., 2024), are ill-suited for this purpose. This is especially important for entity arguments; most EE systems either do not link entities, or use tools that link to Wikidata (Wen et al., 2021) or Wikipedia (Li et al., 2019a; 2020a), which do not necessarily match the expectations of the domain, leading to the need for domain-specific entity datasets and systems (Wei et al., 2016). As such, an EE system should facilitate event argument normalization, and support linking entities to a any provided entity database.

**High Demand for Annotation Quality** Even manual annotation of events is challenging, and poor annotation quality is especially detrimental as it contributes to biased inferences in high-impact policy decisions such as international peacemaking efforts (Andrea Ruggeri & Dorussen, 2011). This often necessitates expert annotations instead of crowdsourcing (Raleigh et al., 2010; Caselli & Huang, 2012). As such, automatic EE systems should be built and evaluated using high quality data.

**Multilinguality**  To study the real world, we often need a *global* view, which necessitates support for a wide range of languages, especially low-resource ones, as for example much of the political analysis of outbreaks and conflicts is focused on the global south and the international setting. Existing event datasets only cover a few languages such as English and Chinese, therefore, EE systems are not properly evaluated on how well they can be used to study global phenomena.

**Flexible Schema and Ontology**  It is important to support custom schemas and entity lists. Many codebooks have been developed for events over many decades of work (Azar, 1980; McClelland, 1978; Walker et al., 2006; Gerner et al., 2008; Walker et al., 2006; Halterman et al., 2023a; Tracey et al., 2022; Duruşan et al., 2022). Oftentimes, scholars define a new domain-specific schema for the phenomena they want to study de Mesquita et al. (2015). While recent work in zero-shot information extraction has made advancements in this direction Sainz et al. (2024), they do not generalize well to arbitrarily varied schemas (Section 5.2).

**Intermediate Annotations are not Available**  In EE literature, the task, datasets and systems are all typically divide into several parts (Huang et al., 2024), each requiring cumbersome span-level annotations: 1) event trigger identification, 2) event trigger classification, 3) event argument identification, 4) event argument classification, 5) entity detection, 6) entity coreference resolution 7) entity linking, and 8) event coreference resolution. Different works either work on a subset of these tasks, or lump them together under the names like event detection (1 and 2), or event argument extraction (3 and 4). Even document-level EE (Tong et al., 2022) relies on span-based intermediate annotations for the task. These intermediate annotations add to the cost of obtaining data for a new domain, and make high quality annotations even more challenging.

In summary, automatic EE in the real world remains challenging. To study a new phenomena (or and old phenomena from a new angle), we need high quality data, often multilingual and with normalized entities across different languages. As an example of the level of effort required, Armed Conflict Location and Event Data (ACLED) (Raleigh et al., 2010; 2023) is annotated by a team of 200 researchers from around the globe (Sam Jones, 2022). To make matters more challenging, off-the-shelf tools like entity linkers that work against Wikidata are not applicable to many domains (Wei et al., 2016). These limitations have remained largely unchanged even with the recent use of large language models (LLMs) and in-context learning in EE (Wang et al., 2023; Sainz et al., 2024).

In this paper, we attempt to bridge this gap between the real-world requirements and EE research by making a real event dataset available, and by evaluating the use of NLP technology to assist in real world EE.

**The Abstractive EE (AEE) Problem.**  Aiming to create a useful tool for real-world EE, we formulate the AEE problem. The distinguishing factor in AEE is that it moves away from the surface form of the text, and focuses on grounding events on a predefined ontology like an entity database, or categorical event arguments [1]. We define the AEE problem as follows:

**Definition 1**

We define event extraction codebook $C = (T, \mathcal{D}, S)$ where

- $T$ is the set of possible event types,
- Each $D \in \mathcal{D}$ is a domain such as integers, real numbers, or a set of known entities,
- A list of event signatures $S = [(t_1, a_{1,1}, \ldots, a_{1,n_1}), \ldots]$, where $n_i$ is the number of arguments for event type $t_i$, and $a_{i,j}$ is an argument with domain $D_{i,j} \in \mathcal{D}$

**Definition 2**

The *Abstractive Event Extraction* (AEE) problem is: given codebook $C = (T, \mathcal{D}, S)$ and writing $w \in W$, extract abstractive event $\text{AEE}(w, C) = (t_i, v_1, \ldots, v_{n_i})$ which is the main event Tong et al. (2022) in $w$, where $t_i$ is the $i$th event type in $T$, $v_j \in D_{i,j}$ and $n_i$ is the number of arguments for event type $t_i$ .

---

[1]The term *abstractive* has been used in other NLP tasks like OpenIE (Pei et al., 2023) and summarization (Radev et al., 2002) to refer to the concept of moving away from the surface form.

In the example in Figure 1, $t_i = \text{MobViolence} \in T$, the first two arguments, group_1 and group_2 represent the two sides in the violence, with $D_{i,1}, D_{i,2}$ being the set of all subsets of possible entities from the event database, the third argument is a location, and domains of the last two arguments, $D_{i,4}, D_{i,5}$, are both booleans.

In AEE, we remove the limitation for arguments to be spans, or explicitly mentioned in the text at all. In addition to the abovementioned benefits, this also enables the annotation of *implicit* event arguments. For instance in Figure 1, the higher-level entities like "Dalit Caste Group" require domain-specific knowledge (the caste system in India in this example), which is provided as a descriptions in the entity database.

**The LEMONADE Dataset.** We present an event dataset for the AEE task called LEMONADE (**L**arge **E**xpert-annotated **M**ultilingual **O**ntology-**N**ormalized **A**bstractive **D**ataset of **E**vents). The dataset is extracted from the high-quality data annotated by experts at ACLED. This data has been used by international organizations like The United Nation's International Organization for Migration, The International Rescue Committee and The European Commission for tracking and predicting forced displacements and evaluating humanitarian efforts (ACLED, 2023).

**Solving the AEE Problem** In this paper, we study the following questions:

1. Given a high-quality AEE training dataset, can we perform AEE effectively?

2. It is costly to create a large high-quality AEE training dataset for new domains. Is it possible to create a zero-shot model for AEE?

The contributions of this paper include:

- A new expert-annotated dataset called LEMONADE. It includes $41,148$ events covering 16 languages, including several languages like Indonesian, Burmese and Nepali that were not previously studied for events in an academic setting. LEMONADE has many entities that do not have Wikidata or Wikipedia entries, making it especially challenging and a suitable testbed for zero-shot entity linking systems.

- Our supervised AEE model achieves $71.6\%$ $F_1$ on LEMONADE, establishing a strong baseline.

- We propose ZEST, a novel zero-shot system for AEE. To handle the full complexity of the real-world AEE problem, we decompose the problem into manageable subproblems; of note is the novel zero-shot entity linking component. The zero-shot ZEST achieves $57.2\%$ on the LEMONADE, which is $13.5\%$ better than existing zero-shot baselines.

## 2 RELATED WORK

The task of Event Extraction aims to extract events and their arguments from a given context. The Message Understanding Conferences (MUC) in the 1990s (Grishman & Sundheim, 1996) were one of the first endeavors at building automated EE systems (Anderson et al., 2012). The datasets prepared for MUC pioneered text spans as the unit of some system outputs. Today's EE research is based on the task formulation of the ACE05 project (Walker et al., 2006), which divides the task into subtask at the sentence level with span-based intermediate annotations (Walker et al., 2006). Li et al. (2021b) extended EE to allow for arguments of an event to be from surrounding sentences, and Li et al. (2021b) introduce the concept of "most informative span" for arguments. Tong et al. (2022) introduced the DocEE dataset, where event arguments are scattered across the document, fully realizing EE as a *document-level* task.

EE has been extensively studied in the AI community (Ji & Grishman, 2008; Liao & Grishman, 2011; Chen et al., 2015; Liu et al., 2018; Yang et al., 2019; Zhu et al., 2024b; Ren et al., 2024; Lai, 2022; Li et al., 2022; Zhou et al., 2020). Previous work has employed a variety of approaches including graph-based modeling, which leverages structured relationships within data (Dutta et al., 2021; Lai et al., 2020; Zhang et al., 2020) and language modeling (He et al., 2015; Michael et al., 2018; Li et al., 2019b; Du & Cardie, 2020). Furthermore, joint modeling techniques (Nguyen et al., 2022; Hsu et al., 2022; Zhang & Ji, 2021) sometimes dubbed end-to-end models (Zheng et al.,

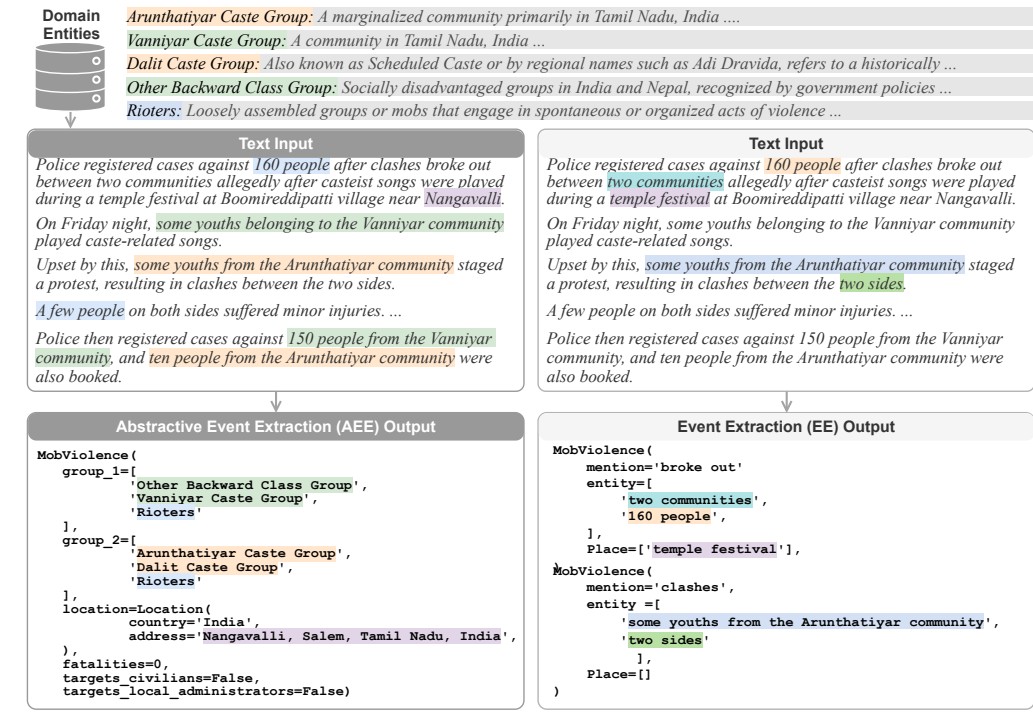

Figure 1: An example of a "Mob Violence" event from the LEMONADE dataset showing the many significant differences between AEE and EE on the same text input. For AEE, entities must be matched to one of the entities in the given domain, whereas EE annotations refer to the entities as span in the text. AEE identifies that it is a single event, whereas EE classifies it as two, with no way to annotate the two sides of the conflict consistently across them. AEE notes the two clashing groups explicitly. Group 1 includes not just "Vanniyar Caste Group" but "Other Backward Class Group", the larger group that the Vanniyar caste belongs to under the Indian government definition, as well as "Rioters" to indicate the presence of rioters. Similarly, group 2 includes not just "Arunthatiyar Caste Group", but the larger group "Dalit Caste Group" and the generic rioters group. The larger group information requires knowledge beyond what is in the text input; this is important to annotate given the known rivalries between the groups. Furthermore, the location information in AEE is much more precise than that of EE, enabling spatial analysis.

2019), integrate multiple EE subtasks to improve extraction accuracy. With the recent advances in generative language models, more research has focused on applying generative methods on event extraction (Shi et al., 2023; Anantheswaran et al., 2023; Li et al., 2021b; Lu et al., 2021), culminating in the use of LLMs (Xu et al., 2023; Wang et al., 2023; Qi et al., 2024). We note that AEE is different from end-to-end approaches, in that it does not rely on intermediate annotations of auxiliary tasks.

Most EE datasets focus on English, and to a lesser extent Chinese (Zhu et al., 2024a; Ren et al., 2024; Walker et al., 2006). Event extraction datasets for other languages include BKEE (Nguyen et al., 2024) for Vietnamese, InDEE-2019 (Maheshwari et al., 2019) for 5 Indic languages, MEE (Pouran Ben Veyseh et al., 2022) for Portuguese, Spanish, Polish, Turkish, Hindi, Japanese and Korean, Zavarella et al. (2014) for Bulgarian, Romanian and Turkish, and Balali et al. (2022) for Farsi.

While there are several socio-political event databases that use automated tools for extraction (Leetaru & Schrodt, 2013; Hallberg, 2012), manual annotation remains the gold standard.

Event extraction in the socio-political domain has long been an important research theme (Raleigh et al., 2010; Chenoweth & Lewis, 2013; Weidmann & Rød, 2019; Kriesi et al., 2019; Hürriyetoğlu et al., 2024b). A line of recent work uses language models to detect socio-political events with nuanced contextual understanding (Tanev, 2024; Tanev & De Longueville, 2023; Mehta et al., 2022; Slavcheva et al., 2023). Since data scarcity is a critical issue in socio-political event extraction, finding innovative data utilization strategies has also become a focus (Loerakker et al., 2024; Bakker et al., 2024; Mutlu & Hürriyetoğlu, 2023; DeLucia et al., 2023; Raj et al., 2022).

## 3   LEMONADE, A MULTILINGUAL AEE DATASET FOR THE REAL WORLD

LEMONADE is an expert-annotated event dataset covering 16 languages: English, Spanish, Arabic, French, Portuguese, Korean, German, Ukrainian, Burmese, Italian, Turkish, Indonesian, Russian, Farsi, Nepali, and Chinese. These languages are selected for their typological diversity (Clark et al., 2020), and span several high and low resource languages. To the best of our knowledge, this is the first event extraction dataset for Burmese, Indonesian and Nepali, and covers the most number of languages than any other event dataset by far.

In event extraction, intermediate annotations like event mentions and entity mentions are expected from datasets and systems (Liu et al., 2021). One event can be mentioned multiple times in the document, and those are called *event coreferences*. LEMONADE, on the other hand, following AEE, does away with annotating entity spans and coreferences, and event mentions and coreferences. Instead, it focuses on actually reporting the event that the document describes.

LEMONADE is based on the Armed Conflict Location and Event Data (ACLED) (Raleigh et al., 2010). Originally published in 2010, ACLED focused on civil war, subnational and transnational violent events in 50 unstable countries, it has since expanded to track more types of political violence event, as well as civil unrest events, in 243 countries and territories in 100 languages in near real-time (Sam Jones, 2022; ACLED, 2023). We chose this as our data source because in addition to the wide language coverage, it has high-quality expert annotations, mitigating quality issues present in many NLP datasets (Campagna et al., 2022).

In the rest of this section, we describe the process of creating LEMONADE.

We preprocess the ACLED data with the goal of transforming it into a format that is more amenable for AI models, while keeping as much of the information as possible. The main challenge is to ensure annotations only contain information that can be extracted or inferred from the input. The steps taken involve data cleaning and reannotation of certain event arguments. The general process was automated as much as possible, and involved spot-checks and several rounds of improvements from two authors of this paper.

**ACLED Annotation and Review Process.**   We start from the publicly available expert annotations of ACLED. ACLED annotations are done by a group of around 200 experts and is updated on a weekly basis. It sources *writings* from news media, international organizations, NGO and security reports, and local partner organizations and select social media channels. It annotates one event per writing, the *main* event excluding historical events that are typically mentioned in writings to provide more context. These writings go through a multi-step review and quality assurance process (ACLED, 2020). The annotation of events is done at a regional level (e.g. the Middle East, Africa etc.) by experts of those regions. These experts have local language skills and knowledge about regional conflicts, and many live within the country they cover. The annotations are then merged by a research manager who reviews these data for inter-coder reliability across the region. Researchers use an annotation tool that provides them with the up-to-date list of entities and locations, and communicate with each other to clarify difficult annotation decisions. After merging regional data, another round of manual reviewing is performed by another expert.

There are 25 politically significant event types covering battles, protests, riots, violence against civilians, political agreements, arrests and more. Appendix D shows the full list of event types and the arguments of each one.

**Data Filtering and Cleaning**   We obtain all events from the first 7 months of 2024. Overall, this includes $112,885$ events, each paired with a writing and an annotated event. After analyzing the data, we realized that many social media posts in the data are accompanied by an image (e.g. protest fliers), and the text alone is not enough to annotate the event. Therefore, we exclude social media posts. We also remove the $1\%$ longest and shortest writings, because very short ones (often from local partner organizations' reports) do not include enough context for annotating the event, and very long texts are often a combination of multiple news articles. This leaves us with 90,035 events sourced mainly from mostly news articles. A number of ACLED events include multiple writings and annotations, each one covering one aspect of the event, for example, a national protest that occurs in multiple cities. We keep one of each event, and are left with $63,217$ events. We further

limit the data to languages that have at least 500 events. We obtain the writings from the provided URLs, and clean them by removing advertisements etc. using an LLM prompt.

**Entity Database**    ACLED annotates entities involved in each event. We provide a database of 6217 entities that appear in ACLED events in 2024. In each event, entity arguments have a small subset of this database as their value. This list contains both generic entities (Halterman et al., 2023b) like "Rioters", "Women", "Students", and specific entities like "Vanniyar Caste Group".

Often, domain knowledge is required for entity linking in specialized domains. The example in Figure 1 demonstrates this. There are entities that are explicit mentioned in the source article and need to be linked to the database, and there are entities whose role in the event is *implicit*, or are annotated because of their relationship with an explicit entity.

While it is possible to learn entities of a domain with enough data, we want LEMONADE to enable research on zero-shot entity linking in this challenging setting. Therefore, to make domain knowledge available to models in a realistic way, we also provide a one-paragraph description for each entity. These descriptions are meant to provide entity linking models with enough context and domain knowledge to understand domain entities, especially the long tail (Mallen et al., 2023).

**Location.**    Location is a crucial event argument for conflict events. In ACLED, the country and up to three subnational administrative levels are annotated (ACLED, 2023). In cases where an event happens in an unknown location within a larger geographic area, or near a city or border, the closest location is used as the location. In rare cases, other sources like maps are used to pinpoint the exact location of an event. There are two issues with this approach when used for building or evaluating AI models. First, because the annotations contain locations that are not inferrable from the writing, this would encourage models to *hallucinate* a location. Second, it puts the burden of knowing the location hierarchy (e.g. which town is in which province) on the shoulders of the AI model. For these reasons, we provide a simpler definition for location, and reannotate the location argument to match this definition: "*The location argument is the most specific place that is supported by the writing*".

For reannotation, we use the original ACLED location annotations to consult the OpenStreetMap geographic database (OpenStreetMap contributors, 2017) to find the full hierarchy of location above the neighborhood level for each event. We then start from the lowest location level and remove the items that are not supported by the writing, until we reach one that is. We then keep that location and all levels above that. A carefully designed LLM prompt was used for this last stage. The final location arguments were spot-checked by the authors, and 97% of them were correct according to the above definition. The Location argument in Figure 1 shows an example output of this reannotation process. In addition, during evaluation (Section 5.1), we first use the same geographic database to normalize locations, in case the AI model predictions have slight differences such as different spelling of town names, or a missing province name when the town name is extracted correctly.

**Schematization**    ACLED uses the same event argument roles for all event types, resulting in some argument roles being always empty for some event types or the argument names being too generic. we define separate event argument roles for each event type. For example, we remove "fatalities" argument from "Peaceful Protest" and rename "actor 1" to "Abductor" for the "Abduction or forced disappearance" event type. We also provide a short description for each event type, and expert descriptions for each event argument, to facilitate the development of zero-shot models.

Following the recent trend in event extraction, we use Python code to represent annotations. This has been shown to improve the performance of various supervised (Sainz et al., 2024) and few-shot (Wang et al., 2023) models because it makes the labels closer to the code data many language models have been pre-trained on. Furthermore, this enables the use of constrained decoding (Rabinovich et al., 2017; Willard & Louf, 2023) algorithms to eliminate malformed outputs. Appendix D presents the full schema for LEMONADE.

**Data splits**    We provide validation and test sets in 16 languages, and a large training set in English. The data split is across time, meaning that the events in the training set are from the first 6 months of 2024, and the events in validation and tests sets are from July 2024. This mimics the real-world setting where the distribution of events and entities might change over time. Because of this

Table 1: LEMONADE statistics per language.

|        | Total | en    | es   | ar   | fr   | pt   | ko   | de   | uk   | my  | it  | tr  | id  | ru  | fa  | ne  | zh  |
|--------|-------|-------|------|------|------|------|------|------|------|-----|-----|-----|-----|-----|-----|-----|-----|
| Train  | 17000 | 17000 | -    | -    | -    | -    | -    | -    | -    | -   | -   | -   | -   | -   | -   | -   | -   |
| Dev    | 12074 | 1000  | 1000 | 1000 | 1000 | 1000 | 1000 | 1000 | 842  | 724 | 721 | 714 | 703 | 395 | 387 | 316 | 272 |
| Test   | 12074 | 1000  | 1000 | 1000 | 1000 | 1000 | 1000 | 1000 | 842  | 724 | 721 | 714 | 703 | 395 | 387 | 316 | 272 |

split, 22.1% of entities in the validation and test sets are not seen in the training set. The split between validation and test sets is random. Table 1 shows the language statistics of LEMONADE, and Appendix A contains event type and geographical distribution of the dataset.

## 4    ZEST: A ZERO-SHOT AEE MODEL

LEMONADE is the rare case where a large high quality training set is available, but that is not the case for many scenarios. In this paper we want to leverage LEMONADE to understand how we can tackle AEE, without requiring expert annotations for training. We assume no access to training data in any language, and that only the information about the schema and the domain is provided in the form the event ontology, and the entity database.

For this, we present a zero-shot system called ZEST. ZEST uses zero-shot in-context learning (i.e. only instructions). The inputs to AEE, writing $w$ and codebook $C$, can be really long, with each event type having its argument signature. It is ineffective, if we present the LLM with the entire codebook. Our preliminary experiments showed that adding few-shot examples is inadequate, perhaps also due to the large size of $w$.

To address the complexity of AEE, we break it down into 3 simpler tasks that are more amenable to in-context learning:

1. *Event Detection* (ED) finds the abstractive event type;

2. *Abstractive Entity Detection and Linking* (EDL) finds a subset of the entity database involved in the abstractive event and assign them to the correct event argument;

3. *Abstractive Event Argument Extraction* (EAE) finds the event arguments for non-entity arguments, given the event type.

Note that EDL and EAE are handled differently from each other in ZEST, because the very large size of the entity domain adds more challenges that a zero-shot system needs to handle. Formally:

**Definition 3**

Given codebook $C = (T, \mathcal{D}, S)$ and writing $w \in W$,

$$\text{ED} = t, \text{where AEE}(w, C) = (t, \ldots)$$

$$\text{EDL}(w, C, t) = V, \text{where AEE}(w, C) = (t, v_1, \ldots) \text{ and } v_i \in V$$

$$\text{EAE}(w, C, t) = [v_1, \ldots], \text{where AEE}(w, C) = (t, v_1, \ldots)$$

**ZEST Event detection (ED)**    Given that the list of event types ($T$) is relatively small (25 in the case of LEMONADE), event detection can be done as a zero-shot in-context learning task. The prompt (Table 5) includes the input writing $w$ and a list of event types and their descriptions. The task is to return the most likely event type $t$. We use chain-of-thought (Wei et al., 2023) for this prompt.

**ZEST Entity Detection and Linking (EDL)**    Once the event type is determined, the next step is to narrow down the list of possible entities that are involved in the event.

We found that in-context learning cannot handle the large number of entities (6217 in the case of LEMONADE) in the AEE task if they are presented in one prompt. Therefore, we tackle this in two stages: the first narrows down the number of candidate entities and the second stage further more closely filters down the set.

We divide the list of all possible entities into groups of $N$. We use a simple zero-shot prompt (Table 6) that given $w$, $t$ and all entities in each group, removes the irrelevant entities. In practice, we find that a wide range of values for $N$ works well, and we choose $N = 63$ (i.e. 100 groups in the case of LEMONADE) in our experiments.

Given the $w$ and the smaller list of entities and their description, the next step uses another prompt (Table 7) to find evidence of each entity's involvement in the event and remove the ones for which we cannot find any evidence. We find that this formulation is especially helpful in identifying implicit entities.

The last step is to match each entity with its correct event argument (e.g. is an entity the "victim" or the "perpetrator" of an "Attack" event?). For this, we use another prompt (Table 8) that given a list of entities and event arguments, outputs a mapping between the two.

**ZEST Event Argument Extraction**   Given the identified event type and entity arguments, we now extract all the other arguments using an approach similar to Wang et al. (2023). This is done using a prompt (Table 9) that given $w$ and the event type signature for $t$, outputs all non-entity argument values.

## 5   EXPERIMENTS AND RESULTS

In addition to the zero-shot setting, we also measure the performance of the system separately in English and non-English languages (i.e. zero-shot cross-lingual generalization), and unseen actors (i.e. zero-shot generalization to unseen actors).

### 5.1   METRICS

To evaluate a predicted event against a gold event from LEMONADE, we first normalize the location arguments using a lookup in the OpenStreetMap geographic database. We then use simple string equality to calculate precision, recall and micro-averaged $F_1$ (Manning et al., 2008).

For ED, we compare the predicted event type against the gold event type, and report the micro averaged **ED** $F_1$. For EAE, we force the gold event type as the first part of the model output, and have it generate event arguments and their values $\{(a'_1, v'_1), ...\}$. We then consider this set as the returned result, and calculate its precision, recall and $F_1$ against the gold $\{(a_1, v_1), ...\}$ and report **EAE** $F_1$. In other words, two arguments are considered equal if their argument *and* values match.

We define and choose **AEE** $F_1$ as our main metric, which is similar to EAE $F_1$, except that if the predicted event type is incorrect, all arguments are considered incorrect, contributing to both false positives and false negatives in the calculation of $F_1$.

For entities, we report **EDL** $F_1$, which is the result of comparing the entity IDs between prediction and gold. Note that EDL $F_1$ ignores the argument type $a$. Additionally, we report EDL on two interesting subsets of entities: entities that have been *seen* in LEMONADE's training set, and those that are *unseen*.

### 5.2   SETUP

**Supervised Setting**   If enough training data is available, we show that simply modeling the task as a sequence-to-sequence task is effective: the model is given $w$ as input, and is trained to predict the full Python code representing the event. For this setting, we fine-tune several language models on the English LEMONADE training set. We use the 8-billion parameter version of LLaMA 3.1 (Dubey et al., 2024) for its strong performance in multilingual benchmarks. We also include LLaMAX (Lu et al., 2024), which extends LLaMA 3 to more than 100 languages by continual pre-training and the 12B parameter model `Mistral-Nemo-Base-2407` for its tokenizer's better support of non-Latin scripts. For comparison, we also include the 7-billion parameter version of LLaMA 2 (Touvron et al., 2023), which has not been specifically trained for non-English languages, though its pre-training data contains a small amount. The base (non-instruction-tuned) versions of all models are used.

We also experiment with *translation at test time* (Moradshahi et al., 2020), by translating all $w$ in the test/dev sets into English using GPT-4o. This way, the supervised AEE model receives English text as input at inference time, which matches its training data more closely.

**Zero-shot Setting** For all zero-shot experiments, we use GPT-4o version `gpt-4o-2024-08-06`. We measure the impact of the zero-shot EDL of ZEST separately. We use constrained decoding when generating Python code for all settings, so the outputs are always syntactically valid, e.g. the event arguments are valid for the predicted event type. The most promising zero-shot baseline from the EE literature is GoLLIE (Sainz et al., 2024), given that it claims to support flexible schemas. However, while we were able to reproduce its results on the datasets they experimented with, the outputs were poor when evaluated with even a small change to the "Location" field. We believe this is due to the limited diversity in event schemas in its training data.

## 5.3 OVERALL RESULTS

Table 2 shows the result of our supervised and zero-shot systems on the LEMONADE test set, averaged over the 16 languages. LLaMA 3.1, LLaMAX and Mistral NeMo perform similarly, all within $0.2\%$ of each other in the AEE $F_1$ metric. The added language support in LLaMAX has minimal effect. We attribute this to the fact that in LEMONADE, all outputs are normalized (and therefore in English), so the models have an easier task generalizing to new languages. Translating the documents to English, improves the AEE $F_1$ between $1.9\%$ and $4.3\%$. The LLaMA 2 model which has not gone through special multilingual pre-training, on the other hand, sees the most benefit from *translation at test time*, with an improvement of $9.5\%$ in AEE $F_1$.

As for our zero-shot system ZEST, it is $14.4\%$ and $10.5\%$ behind the best supervised (Mistral Nemo + translation) and the best supervised model without translation (LLaMAX) in terms of AEE $F_1$. The majority of this gap comes from ED ($9.8\%$ and $8.3\%$ gap), while EAE is closer ($8.2\%$ and $5.0\%$ lower). One area that ZEST shines, is in entity linking accuracy. Specifically, it adds $45.5\%$ over the baseline of directly generating entities with LLM, and outperforms the supervised models in the unseen entity setting by at least $32.6\%$ When training data is available for entities, however, supervised models significantly outperform ZEST.

Table 2: Results of our zero-shot and supervised systems on the test set of LEMONADE. Numbers are averages over all 16 languages. The highest number for each metric is in **bold**.

| | ED $F_1$ | EAE $F_1$ | AEE $F_1$ | EDL $F_1$ (all) | EDL $F_1$ (seen) | EDL $F_1$ (unseen) |
|---|---|---|---|---|---|---|
| Supervised Models | | | | | | |
| LLaMA 3.1 (8B) | 87.3 | 77.3 | 67.5 | 68.6 | 80.9 | 14.1 |
| + *translation at test time* | 88.5 | **80.2** | 71.0 | 69.9 | 82.0 | 17.2 |
| Mistral NeMo (12B) | 87.9 | 76.6 | 67.3 | 69.2 | 81.5 | 12.1 |
| + *translation at test time* | **89.6** | 79.9 | **71.6** | **71.3** | **83.0** | 17.7 |
| LLaMAX (8B) | 88.3 | 76.7 | 67.7 | 68.3 | 80.5 | 13.3 |
| + *translation at test time* | 88.1 | 79.0 | 69.6 | 70.3 | 82.2 | 16.3 |
| LLaMA 2 (7B) | 82.1 | 73.3 | 60.2 | 64.2 | 75.6 | 11.3 |
| + *translation at test time* | 88.0 | 79.2 | 69.7 | 69.3 | 80.5 | 17.2 |
| Zero-shot Models | | | | | | |
| ZEST | 79.8 | 71.7 | 57.2 | 54.0 | 55.3 | **50.3** |
| - *entity linking* | 79.8 | 54.8 | 43.7 | 8.5 | 18.4 | 0.2 |

## 5.4 PER-LANGUAGE RESULTS

We take a closer look at the performance of the best cross-lingual model (LLaMAX without translation), and ZEST in each individual language. Table 3 shows per-language results on the LEMONADE test set. The largest gap between the supervised model and ZEST is in English ($25\%$ in AEE $F_1$), which is reasonable given the training data for LLaMAX is in English. We provide the language acronym mapping in Appendix.

Our analysis of the outputs show that the variance between languages is mainly due to the different distributions of event types. For instance, in politically stable countries (where writings in Korean, Italian, Chinese and German languages come from), almost all event types are of "Protest" type, and there are no battles or remote violence reported, and we observe that ED score for supervised methods is really high. Overall, given the abstractive nature of the task, and the fact that the gold annotations are normalized and in English, the effect of cross-lingual capabilities of the model becomes less influential relative to extractive EE.

ZEST outperform the supervised model in Burmese (my). This language, widely spoken in Myanmar, has a wide range of event types in LEMONADE, and due to its low-resource nature, is quite challenging in the cross-lingual setting. Russian, Farsi, Turkish and French are other languages where the gap is relatively small.

Table 3: AEE $F_1$ of two models on individual languages of the LEMONADE test set.

| Model | en | es | ar | fr | pt | ko | de | uk | my | it | tr | id | ru | fa | ne | zh |
|-------|-----|-----|-----|-----|-----|-----|-----|-----|-----|-----|-----|-----|-----|-----|-----|-----|
| LLaMAX | 76.7 | 72.3 | 48.9 | 65.6 | 66.1 | 81.3 | 78.5 | 62.9 | 41.2 | 76.6 | 64.0 | 76.4 | 63.7 | 67.7 | 65.9 | 79.3 |
| ZEST | 51.7 | 60.3 | 40.9 | 61.5 | 52.4 | 57.4 | 71.0 | 54.7 | 43.8 | 70.4 | 60.0 | 60.1 | 61.4 | 63.8 | 51.9 | 67.0 |

## 6 CONCLUSIONS

This paper introduces the task of abstractive event extraction (AEE), which more closely matches the requirements of event extraction for real-world applications. We have derived a large high-quality dataset for the AEE task, in 16 different languages, from the expert-annotated data created by ACLED.

We introduced ZEST, a novel zero-shot AEE system, that achieves 57.2%. With our supervised model that achieves 71.6% $F_1$, they represent strong baselines for this new dataset.

Reaching 71.6% with supervised learning, our system can be helpful to human annotators by providing them with the first draft to accelerate the annotation task. Furthermore, errors do occur in human-annotated data. The automatically generated results can be used to double check human annotations. During the error analysis of ZEST for example, we discovered a few missing entities in the manual annotations. In contrast, we note that the original EE formulation that refers to entities as spans in the text is not useful for event analysts, nor can it be used to help human annotators.

## ETHICS STATEMENT

No human subjects were involved in this study. We will release LEMONADE in accordance with the ACLED Terms of Use. ACLED data do not contain personally identifiable information (e.g. names of individuals or mobile device IDs), and cannot be used to track individuals. No crowdsourcing was performed as part of this paper.

## REPRODUCIBILITY

Appendix C contains the hyperparameters of all fine-tuned models. Section 5.2 includes more details on the specific models and LLMs used. All LLM prompts used in ZEST are listed in Appendix B.

Section 5.1 explains the metrics used. We provide a detailed description of the preprocessing steps of the LEMONADE dataset in Section 3, and its statistics in Section 3 and Appendix A.

We are also attaching an anonymized version of our code for ZEST, and a sample of the LEMONADE dataset to this submission. We will publicly release the code and the full dataset upon publication.

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

## A    LEMONADE STATISTICS

Tables 4 and Figure 2 show the distribution of event types and country-level locations of events in LEMONADE respectively.

Table 4: The number of event types in all splits of LEMONADE. While imbalanced, the distribution of event types tracks that of the real world. For example, by far the most common among these events are peaceful protests.

| Event Type | Count |
|---|---|
| GovernmentRegainsTerritory | 6 |
| NonStateActorOvertakesTerritory | 55 |
| ArmedClash | 2775 |
| ExcessiveForceAgainstProtestors | 30 |
| ProtestWithIntervention | 993 |
| PeacefulProtest | 24805 |
| ViolentDemonstration | 910 |
| MobViolence | 2015 |
| AirOrDroneStrike | 1218 |
| SuicideBomb | 4 |
| ShellingOrArtilleryOrMissileAttack | 1161 |
| RemoteExplosiveOrLandmineOrIED | 480 |
| Grenade | 93 |
| SexualViolence | 54 |
| Attack | 3231 |
| AbductionOrForcedDisappearance | 304 |
| Agreement | 68 |
| Arrest | 631 |
| ChangeToArmedGroup | 362 |
| DisruptedWeaponsUse | 641 |
| BaseEstablished | 12 |
| LootingOrPropertyDestruction | 780 |
| NonViolentTransferOfTerritory | 19 |
| OtherStrategicDevelopment | 500 |

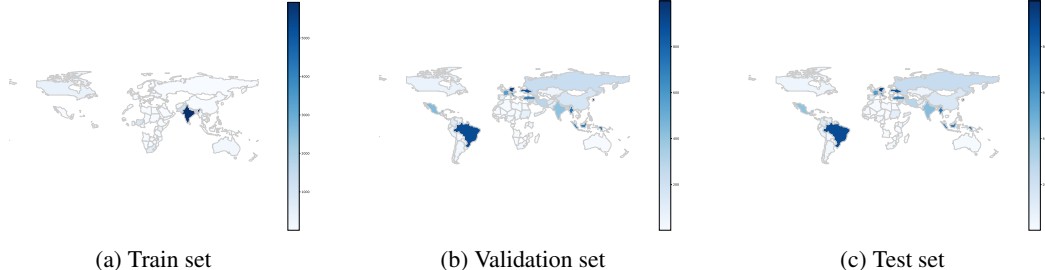

| (a) Train set | (b) Validation set | (c) Test set |

Figure 2: Distribution of event locations in LEMONADE. Note that the dataset includes more specific locations, but here we only plot the country level. In addition to being linguistically diverse, LEMONADE is also geographically diverse. The distribution of the train set is skewed towards India, because it only contains English events.

# B ALL PROMPTS FOR ZEST

Here we provide the prompts used in ZEST. Some prompts are edited for brevity. The full text of prompts can be obtained from our code. The syntax used is the Jinja2 template language, which supports Python-like loops (`{% for %}{% endfor %}`), conditions (`{% if %}{% endif %}`), variables (`{{ var }}`) and comments (#).

```
# instruction
You are tasked with determining the best matching Event types for a given
    news article. You will be provided with annotation guidelines and a
    news article to analyze. Your goal is to identify the most relevant
    event types and rank them in order of their match to the article
    content.

# input
Here is the news article you need to analyze:
{{ article }}

Now, carefully review the annotation guidelines for various event types:

{% for ed in event_definitions.items() %}
[{{ loop.index }}] "{{ ed[0] }}": {{ ed[1] }}

{% endfor %}

1. For each event type, determine how well it matches the article content
    . Consider the following factors:
   - How closely the event description aligns with the main focus of the
     article
   - The presence of key actors or entities mentioned in the event type
     description
   - The occurrence of specific actions or outcomes associated with the
     event type

2. Rank the event types based on their relevance to the article content.
    Only include event types that have a meaningful connection to the
    article.

3. Output your results using the following format:
   - List the relevant event types in descending order of match quality
   - Use the ">" symbol to separate the event types

Your output should look like this:

[Explain your reasoning for the event types you decide to include, and
    their order]

event_type_1 > event_type_2 > ...

Remember to exclude any event types that are not relevant to the article
    content. Provide only the ranked list of event types in your final
    answer.
```

Table 5: Prompt for event type detection (ED).

```
# instruction
Your task is to select all entities involved in a news article from a
    provided list. An entity is an individual, group, or organization
    involved in an event. This includes:
 - Organized armed groups with political purposes
 - Named entities
 - General terms describing participants like "Rioters", "Protestors", "
   Civilians", "Labour Group", etc.

# input
News article:
{{ article }}

The event you should focus on is the {{ event }} event, which happened in
    {{ country }}.

Guidelines:
1. Read the entire article carefully.
2. Identify groups, organizations, and individuals involved in the
    described events.
3. Note both specific names and generic terms used for participants.
4. Consider entities that may be implicitly involved.
5. For politicians, include the name of their political party or group as
    well, if available in the entity list.
6. Include both specific and generic entities when applicable (e.g., a
    political party leading a protest should be counted as two entities:
    the party name and "Protestors"), if available in the entity list.
7. Include characteristics like ethnicity or religion as separate
    entities when mentioned (e.g., "Latin American Group" or "Women"), if
    available in the entity list.
8. Err on the side of inclusion if unsure about an entity's involvement.

From the following list, select entities involved in the event described
    in this news article:

{% for entity in potential_entities %}
[{{ loop.index }}] {{ entity }}
{% endfor %}

Provide your answer listing one entity name per line:
entity name 1
entity name 2
...
```

Table 6: Prompt for the first stage of Entity Detection and Linking (EDL).

```
# instruction
In this task, an "entity" refers to an individual, group, or entity
    involved in the event described in the news article. entities can
    include:

1. State forces
2. Rebels
3. Militias
4. Identity groups
5. Demonstrators
6. Civilians
7. External or other forces

Most entities in political violence events are organized armed groups
    with a political purpose. They use violence for political means and
    are typically named entities. However, entities can also include
    unorganized groups like rioters, as well as protestors and civilians.

Your task is to find supporting evidence for each of the specified
    entities in the given article.

# input
Follow these steps carefully:

1. First, you will be provided with the full text of the news article:

{{ article }}

2. Next, you will be given a list of entities involved with the {{
    event_type }} event to search for:

{% for e in entities %}
    {{ e }}
{% endfor %}

3. Identify all supporting evidence of each given entity. These could be
    spans involving:
    - The exact entity name or variations of its name
    - Descriptive phrases that identify the entity
    - Phrases that could be used to infer the involvement of the entity

4. If there are multiple evidence for the involvement of an entity,
    output all of them.

5. For each evidence you find for an entity, provide your answer in the
    provided structure.
    Notes:
    - Include the original entity name in the `entity_original` field to
    denote which entities the evidence is for.
    - The character index starts at 0 for the first character of the
    article.
    - If there are multiple evidences for an entity, provide multiple `
    entitiespan`s for it.
    - If no evidence is found for an entity, respond with a mostly empty `
    entitiespan` and only fill the `explanation` field.

Remember to be precise in your span detection and provide clear
    explanations for each evidence span.
```

Table 7: Prompt for the second stage of Entity Detection and Linking (EDL).

```
# instruction
An "entity" refers to an individual, group, or entity involved in an
    event. Most entities in political violence events are organized armed
     groups with a political purpose. They use violence for political
    means and are typically named entities. However, entities can also
    include unorganized groups like Rioters, Protestors and Civilians.
An entity can be a generic term like "Students" or "Protestors", or a
    specific political group, militia, or armed group
Never use an individual's name as an entity. If a politician is mentioned
    , use the name of the political party or group they belong to.
Sometimes, a specific entity is accompanied by a generic entity. For
    example, a political party leading a protest should be counted as two
     actors: the political party, and "Protestors".
You will be given a news article, an event extracted from it, and a list
    of actors. Your task is to assign each entity to the correct event
    argument based on the information provided in the news article.

# input
First, carefully read the following news article:
{{ article }}

Now, consider the following event extracted from the article:
{{ event }}

Here is the list of actors to be assigned to event arguments:
{% for e in entities %}
- {{ e }}
{% endfor %}

You need to assign each actor to one of the following event arguments. Do
    not modify any other part of the event.
{% for field in actor_fields %}
- {{ field }}
{% endfor %}

To complete this task, follow these steps:

1. Analyze the news article and the extracted event carefully.
2. For each actor in the provided list, determine their role in the event
    based on the information in the news article. Note that some actors
    may not be involved in the event at all, in which case, simply ignore
     those.
3. Assign each actor to the most appropriate event argument.
4. If an event argument doesn't have a corresponding actor, leave it as
    an empty list.

Output the completed event arguments with the assigned actors in the
    given JSON format. Note that you should always include the full name
    of the actor.
```

Table 8: Prompt used for assigning entities to their correct event argument. A Pydantic schema is also given to the model to follow.

```
# instruction
You are an AI assistant tasked with extracting event arguments from a
    given news article. You will be provided with annotation guidelines
    for an event type and a news article to analyze.

# input
{{ article }}.
Extract the arguments of the main event in this article, which is of type
    {{ event_type }}.
For "entity" arguments, note that an entity can be a generic term like "
    Students" or "Protestors", or specific political groups, militia,
    armed groups, etc. Never use an individual's name as an entity.

Sometimes, a specific entity is accompanied with a generic entity. For
    example if a political party is leading a protest, both the political
    party's name and the "Protestors" should be included as entities.
When identifying an entity, provide as much information as possible.
```

Table 9: Prompt used for EAE.

## C    HYPERPARAMETERS

All fine-tuned models were fine-tuned with batch size 128 for 5 epochs. The final model check-point was selected for evaluation. Learning rate of $2 \times 10^{-5}$, cosine learning rate scheduler and AdamW (Loshchilov & Hutter, 2017) optimizer are used.

Training is done on a machine with 4 NVIDIA A100 GPU with 80GB, using DeepSpeed (Rasley et al., 2020) and the Transformers (Wolf et al., 2019) library.

For access to GPT-4o model, we used the OpenAI API. For access to OpenStreetMap, we used the publicly hosted version via Nominatim https://nominatim.openstreetmap.org/

## D    FULL SCHEMA OF LEMONADE

The following is the full schema of LEMONADE, after conversion to Python code, in Pydan-tic (Colvin et al., 2024) format. Abstract classes (denoted by ABC are only meant to group event types together and store common event arguments, are not counted as an event type, and are not used by ZEST. Docstrings are modified from the ACLED codebook (ACLED, 2023). WomenTargetedCategory and Location are two event types.

```
class Battle(ACLEDEvent, ABC):
    """
    A "Battle" event is defined as a violent interaction between two organized armed groups at a particular
      time and location. "Battle" can occur between armed and organized state, non-state, and external groups,
      and in any combination therein. There is no fatality minimum necessary for inclusion. Civilians can be
      harmed in the course of larger "Battle" events if they are caught in the crossfire, for example, or
      affected by strikes on military targets, which is commonly referred to as "collateral damage" (for more,
      see Indirect Killing of Civilians). When civilians are harmed in a "Battle" event, they are not
      recorded as an "Actor", nor is a separate civilian-specific event recorded. If any civilian fatalities
      are reported as part of a battle, they are aggregated in the "Fatalities" field for the "Battle" event.
    The specific elements of the definition of a "Battle" event are as follows:
    Violent interaction: the exchange of armed force, or the use of armed force at close distance, between
      armed groups capable of inflicting harm upon the opposing side.
    Organized armed groups: collective actors assumed to be operating cohesively around an agenda, identity,
      or political purpose, using weapons to inflict harm. These groups frequently have a designated name and
      stated agenda.
    The "Battle" event type may include: ground clashes between different armed groups, ground clashes between
      armed groups supported by artillery fire or airstrikes, ambushes of on-duty soldiers or armed militants
      , exchanges of artillery fire, ground attacks against military or militant positions, air attacks where
      ground forces are able to effectively fire on the aircraft, and air-to-air combat.
    Cases where territory is regained or overtaken without resistance or armed interaction are not recorded as
      "Battle" events. Instead, they are recorded as "NonStateActorOvertakesTerritory" under the "
      StrategicDevelopment" event type
    "Battle" event type has the following subtypes:
    - GovernmentRegainsTerritory: Government forces or their affiliates regain control of a location from
      competing state forces or non-state groups through armed interaction.
    - NonStateActorOvertakesTerritory: A non-state actor or foreign state actor captures territory from an
      opposing government or non-state actor through armed interaction, establishing a monopoly of force
      within that territory.
```

```
- ArmedClash: Armed, organized groups engage in a battle without significant changes in territorial
  control.
"""

location: Location = Field(..., description="Location where the event takes place")
fatalities: Optional[int] = Field(
    ...,
    description="Total number of fatalities, if known",
)

class GovernmentRegainsTerritory(Battle):
    """
    Is a type of "Battle" event. This event type is used when government forces or their affiliates that are
      fighting against competing state forces or against a non-state group regain control of a location
      through armed interaction. This event type is only recorded for the re-establishment of government
      control and not for cases where competing non-state actors exchange control. Short-lived and/or small-
      scale territorial exchanges that do not last for more than one day are recorded as "ArmedClash".
    """

    government_force: List[str] = Field(
        ...,
        description="The government forces or their affiliates that regain control of the territory",
    )
    adversary: List[str] = Field(
        ...,
        description="The competing state forces or non-state group that lose control of the territory. Can be
        State Forces, Rebel Groups, Political Militias, Identity Militias or External Forces",
    )

class NonStateActorOvertakesTerritory(Battle):
    """
    Is a type of "Battle" event. This event type is used when a non-state actor (excluding those operating
      directly on behalf of the government) or a foreign state actor, through armed interaction, captures
      territory from an opposing government or non-state actor; as a result, they are regarded as having a
      monopoly of force within that territory. Short-lived and/or small-scale territorial exchanges that do
      not last for more than one day are recorded as "ArmedClash" events. In cases where non-state forces
      fight with opposing actors in a location many times before gaining control, only the final territorial
      acquisition is recorded as "Non-state actor overtakes territory". All other battles in that location are
       recorded as "ArmedClash".
    """

    non_state_actor: List[str] = Field(
        ...,
        description="The non-state actor overtaking territory. Can be Rebel Groups, Political Militias,
        Identity Militias or External Forces",
    )
    adversary: List[str] = Field(
        ...,
        description="The opposing government or non-state actor from whom the territory was taken. Can be
        State Forces, Rebel Groups, Political Militias, Identity Militias or External Forces",
    )

class ArmedClash(Battle):
    """
    Is a type of "Battle" event. This event type is used when two organized groups like State Forces, Rebel
      Groups, Political Militias, Identity Militias or External Forces engage in a battle, and no reports
      indicate a significant change in territorial control.
    `side_1` and `side_2` denote the two sides of the armed clash.
    Excludes demonstrations that turn violent, riots, and other forms of violence that are not organized armed
      clashes.
    """

    side_1: List[str] = Field(
        ...,
        description="Groups involved in the clash. Can be State Forces, Rebel Groups, Political Militias,
        Identity Militias or External Forces",
    )
    side_2: List[str] = Field(
        ...,
        description="Groups involved in the clash. Can be State Forces, Rebel Groups, Political Militias,
        Identity Militias or External Forces",
    )
    targets_local_administrators: bool = Field(
        ...,
        description="Whether this violence is affecting local government officials and administrators -
        including governors, mayors, councilors, and other civil servants.",
    )
    women_targeted: List[WomenTargetedCategory] = Field(
        ...,
        description="The category of violence against women, if any. If this violence is not targeting women,
        this should be an empty list.",
    )

class Protest(ACLEDEvent, ABC):
```

```
1566
1567        """
           A "Protest" event is defined as an in-person public demonstration of three or more participants in which
1568        the participants do not engage in violence, though violence may be used against them. Events include
           individuals and groups who peacefully demonstrate against a political entity, government institution,
1569        policy, group, tradition, business, or other private institution. The following are not recorded as "
           Protest" events: symbolic public acts such as displays of flags or public prayers (unless they are
1570        accompanied by a demonstration); legislative protests, such as parliamentary walkouts or members of
           parliaments staying silent; strikes (unless they are accompanied by a demonstration); and individual
1571        acts such as self-harm actions like individual immolations or hunger strikes.
1572      Protestor are noted by generic actor name "Protestor". If they are representing a group, the name of that
           group is also recorded in the field.
1573      "Protest" event type has the following subtypes:
          - ExcessiveForceAgainstProtestors: Peaceful protestor are targeted with lethal violence or violence
1574        resulting in serious injuries by state or non-state actors.
          - ProtestWithIntervention: A peaceful protest is physically dispersed or suppressed without serious
1575        injuries, or protestor interact with armed groups or rioters without serious harm, or protestors are
           arrested.
1576      - PeacefulProtest: Demonstrators gather for a protest without engaging in violence or rioting and are not
           met with force or intervention.
1577
1578        """
1579
           location: Location = Field(..., description="Location where the event takes place")
1580       protestors: List[str] = Field(
1581           ...,
              description="List of protestor groups or individuals involved in the protest",
1582       )
1583

1584    class ExcessiveForceAgainstProtestors(Protest):
1585        """
           Is a type of "Protest" event (Protest events include individuals and groups who peacefully demonstrate
1586        against a political entity, government institution, policy, group, tradition, business, or other private
           institution.) This event type is used when individuals are engaged in a peaceful protest and are
1587        targeted with lethal violence or violence resulting in serious injuries (e.g. requiring hospitalization)
           . This includes situations where remote explosives, such as improvised explosive devices, are used to
1588        target protestors, as well as situations where non-state actors, such as rebel groups, target protestors
           .
1589        """
1590
           # Possible "Interaction" codes include: 16, 26, 36, 46, 56, and 68.
1591
           perpetrators: List[str] = Field(
1592           ...,
              description="Entities perpetrating the violence. Can be State Forces, Rebel Groups, Political Militias
1593        , Identity Militias, External Forces",
           )
1594       targets_civilians: bool = Field(
1595           ...,
              description="Indicates if the 'ExcessiveForceAgainstProtestors' event is mainly or only targeting
1596        civilians. E.g. state forces using lethal force to disperse peaceful protestors.",
           )
1597
1598       fatalities: Optional[int] = Field(
1599           ...,
              description="Total number of fatalities, if known",
1600       )
1601

1602    class ProtestWithIntervention(Protest):
1603        """
           Is a type of "Protest" event. This event type is used when individuals are engaged in a peaceful protest
1604        during which there is a physically violent attempt to disperse or suppress the protest, which resulted
           in arrests, or minor injuries . If there is intervention, but not violent, the event is recorded as "
1605        PeacefulProtest" event type.
           """
1606
1607       perpetrators: List[str] = Field(
1608           ...,
              description="Group(s) or entities attempting to disperse or suppress the protest",
1609       )
           fatalities: Optional[int] = Field(
1610           ...,
              description="Total number of fatalities, if known",
1611       )
1612

1613    class PeacefulProtest(Protest):
1614        """
           Is a type of "Protest" event (Protest events include individuals and groups who peacefully demonstrate
1615        against a political entity, government institution, policy, group, tradition, business, or other private
           institution.) This event type is used when demonstrators gather for a protest and do not engage in
1616        violence or other forms of rioting activity, such as property destruction, and are not met with any sort
           of violent intervention.
1617        """
1618
           # Possible "Interaction" codes include: 60, 66, and 67.
1619
           counter_protestors: List[str] = Field(
```

```
1620
1621            ..., description="Groups or entities engaged in counter protest, if any"
             )
1622
1623
1624    class Riot(ACLEDEvent, ABC):
             """
1625         "Riot" are violent events where demonstrators or mobs of three or more engage in violent or destructive
               acts, including but not limited to physical fights, rock throwing, property destruction, etc. They may
1626           engage individuals, property, businesses, other rioting groups, or armed actors. Rioters are noted by
               generic actor name "Rioters". If rioters are affiliated with a specific group - which may or may not be
1627           armed - or identity group, that group is recorded in the respective "Actor" field. Riots may begin as
               peaceful protests, or a mob may have the intention to engage in violence from the outset.
1628         "Riot" event type has the following subtypes:
             - ViolentDemonstration: Demonstrators engage in violence or destructive activities, such as physical
1629           clashes, vandalism, or road-blocking, regardless of who initiated the violence.
             - MobViolence: Rioters violently interact with other rioters, civilians, property, or armed groups outside
1630             of demonstration contexts, often involving disorderly crowds with the intention to cause harm or
               disruption.
1631
             """
1632
1633         location: Location = Field(..., description="Location where the event takes place")
1634         fatalities: Optional[int] = Field(
                 ...,
1635             description="Total number of fatalities, if known",
             )
1636         targets_civilians: bool = Field(
1637             ...,
                 description="Indicates if the 'Riot' event is mainly or only targeting civilians. E.g. a village mob
1638           assaulting another villager over a land dispute.",
             )
1639         group_1: List[str] = Field(
1640             ..., description="Group or individual involved in the violence"
             )
1641         group_2: List[str] = Field(
1642             ...,
                 description="The other group or individual involved in the violence, if any",
1643         )
             targets_local_administrators: bool = Field(
1644             ...,
                 description="Whether this violence is affecting local government officials and administrators -
1645           including governors, mayors, councilors, and other civil servants.",
             )
1646         women_targeted: List[WomenTargetedCategory] = Field(
1647             ...,
                 description="The category of violence against women, if any. If this violence is not targeting women,
1648           this should be an empty list.",
             )
1649
1650
1651    class ViolentDemonstration(Riot):
             """
1652         Is a type of "Riot" event. This event type is used when demonstrators engage in violence and/or
               destructive activity. Examples include physical clashes with other demonstrators or government forces;
1653           vandalism; and road-blocking using barricades, burning tires, or other material. The coding of an event
               as a "Violent demonstration" does not necessarily indicate that demonstrators initiated the violence and
1654           /or destructive actions.
             Excludes events where a weapon is drawn but not used, or when the situation is de-escalated before
1655           violence occurs.
             """
1656
1657
1658
1659    class MobViolence(Riot):
             """
1660         Is a type of "Riot" event. A mob is considered a crowd of people that is disorderly and has the intention
               to cause harm or disruption through violence or property destruction. Note that this type of violence
1661           can also include spontaneous vigilante mobs clashing with other armed groups or attacking civilians.
               While a "Mob violence" event often involves unarmed or crudely armed rioters, on rare occasions, it can
1662           involve violence by people associated with organized groups and/or using more sophisticated weapons,
               such as firearms.
1663         """
1664
1665
1666    class ExplosionOrRemoteViolence(ACLEDEvent, ABC):
             """
1667         "ExplosionOrRemoteViolence" is defined as events as incidents in which one side uses weapon types that, by
                their nature, are at range and widely destructive. The weapons used in "ExplosionOrRemoteViolence"
1668           events are explosive devices, including but not limited to: bombs, grenades, improvised explosive
               devices (IEDs), artillery fire or shelling, missile attacks, air or drone strikes, and other widely
1669           destructive heavy weapons or chemical weapons. Suicide attacks using explosives also fall under this
               category. When an "ExplosionOrRemoteViolence" event is reported in the context of an ongoing battle, it
1670           is merged and recorded as a single "Battles" event. "ExplosionOrRemoteViolence" can be used against
               armed agents as well as civilians.
1671         "ExplosionOrRemoteViolence" event type has the following subtypes:
             - ChemicalWeapon: The use of chemical weapons in warfare without any other engagement.
1672         - AirOrDroneStrike: Air or drone strikes occurring without any other engagement, including attacks by
1673           helicopters.
```

```
1674
1675          - SuicideBomb: A suicide bombing or suicide vehicle-borne improvised explosive device (SVBIED) attack
                without an armed clash.
1676          - ShellingOrArtilleryOrMissileAttack: The use of long-range artillery, missile systems, or other heavy
                weapons platforms without any other engagement.
1677          - RemoteExplosiveOrLandmineOrIED: Detonation of remotely- or victim-activated devices, including landmines
                and IEDs, without any other engagement.
1678          - Grenade: The use of a grenade or similar hand-thrown explosive without any other engagement.
              """
1679
1680          location: Location = Field(..., description="Location where the event takes place")
              targets_civilians: bool = Field(
1681              ...,
                  description="Indicates if the 'ExplosionOrRemoteViolence' event is mainly or only targeting civilians.
1682            E.g. a landmine killing a farmer.",
              )
1683          fatalities: Optional[int] = Field(
1684              ...,
                  description="Total number of fatalities, if known",
1685          )
              attackers: List[str] = Field(..., description="Entities conducting the violence")
1686          targeted_entities: List[str] = Field(
                  ..., description="Entities or actors being targeted"
1687          )
              targets_local_administrators: bool = Field(
1688              ...,
                  description="Whether this violence is affecting local government officials and administrators -
1689            including governors, mayors, councilors, and other civil servants.",
              )
1690          women_targeted: List[WomenTargetedCategory] = Field(
                  ...,
1691              description="The category of violence against women, if any. If this violence is not targeting women,
1692            this should be an empty list.",
              )
1693
1694
1695
class ChemicalWeapon(ExplosionOrRemoteViolence):
1696      """
      Is a type of "ExplosionOrRemoteViolence" event. This event type captures the use of chemical weapons in
1697        warfare in the absence of any other engagement. ACLED considers chemical weapons as all substances
        listed as Schedule 1 of the Chemical Weapons Convention, including sarin gas, mustard gas, chlorine gas,
1698        and anthrax. Napalm and white phosphorus, as well as less-lethal crowd control substances - such as
        tear gas - are not considered chemical weapons within this event type.
1699      """
1700
1701
1702
class AirOrDroneStrike(ExplosionOrRemoteViolence):
1703      """
      Is a type of "ExplosionOrRemoteViolence" event. This event type is used when air or drone strikes take
1704        place in the absence of any other engagement. Please note that any air-to-ground attacks fall under this
         event type, including attacks by helicopters that do not involve exchanges of fire with forces on the
1705        ground.
      """
1706
1707
1708
class SuicideBomb(ExplosionOrRemoteViolence):
1709      """
      Is a type of "ExplosionOrRemoteViolence" event. This event type is used when a suicide bombing occurs in
1710        the absence of an armed clash, such as an exchange of small arms fire with other armed groups. It also
        includes suicide vehicle-borne improvised explosive device (SVBIED) attacks. Note that the suicide
1711        bomber is included in the total number of reported fatalities coded for such events.
      """
1712
1713
1714
class ShellingOrArtilleryOrMissileAttack(ExplosionOrRemoteViolence):
1715      """
      Is a type of "ExplosionOrRemoteViolence" event. This event type captures the use of long-range artillery,
1716        missile systems, or other heavy weapons platforms in the absence of any other engagement. When two armed
         groups exchange long-range fire, it is recorded as an "ArmedClash". "ShellingOrArtilleryOrMissileAttack
1717      " events include attacks described as shelling, the use of artillery and cannons, mortars, guided
        missiles, rockets, grenade launchers, and other heavy weapons platforms. Crewed aircraft shot down by
1718        long-range systems fall under this event type.  Uncrewed armed drones that are shot down, however, are
        recorded as interceptions under "DisruptedWeaponsUse" because people are not targeted (see below).
1719        Similarly, an interception of a missile strike itself (such as by the Iron Dome in Israel) is also
        recorded as "DisruptedWeaponsUse".
1720      """
1721
1722
1723
class RemoteExplosiveOrLandmineOrIED(ExplosionOrRemoteViolence):
      """
      Is a type of "ExplosionOrRemoteViolence" event. This event type is used when remotely- or victim-activated
1724        devices are detonated in the absence of any other engagement. Examples include landmines, IEDs -
1725      whether alone or attached to a vehicle, or any other sort of remotely detonated or triggered explosive.
        Unexploded ordnances (UXO) also fall under this category.
1726      SVBIEDs are recorded as "Suicide bomb" events, while the safe defusal of an explosive or its accidental
        detonation by the actor who planted it (with no other casualties reported) is recorded under "
1727        DisruptedWeaponsUse".
      """
```

```python
class Grenade(ExplosionOrRemoteViolence):
    """
    Is a type of "ExplosionOrRemoteViolence" event. This event type captures the use of a grenade or any other
      similarly hand-thrown explosive, such as an IED that is thrown, in the absence of any other engagement.
      Events involving so-called "crude bombs" (such as Molotov cocktails, firecrackers, cherry bombs, petrol
      bombs, etc.) as well as "stun grenades" are not recorded in this category, but are included under
      either "Riot" or "StrategicDevelopment" depending on the context in which they occurred.
    """

class ViolenceAgainstCivilians(ACLEDEvent, ABC):
    """
    ACLED defines "ViolenceAgainstCivilians" as violent events where an organized armed group inflicts
      violence upon unarmed non-combatants. By definition, civilians are unarmed and cannot engage in
      political violence. Therefore, the violence is understood to be asymmetric as the perpetrator is assumed
       to be the only actor capable of using violence in the event. The perpetrators of such acts include
      state forces and their affiliates, rebels, militias, and external/other forces.
    In cases where the identity and actions of the targets are in question (e.g. the target may be employed as
       a police officer), ACLED determines that if a person is harmed or killed while unarmed and unable to
      either act defensively or counter-attack, this is an act of "ViolenceAgainstCivilians". This includes
      extrajudicial killings of detained combatants or unarmed prisoners of war.
    "ViolenceAgainstCivilians" also includes attempts at inflicting harm (e.g. beating, shooting, torture,
      rape, mutilation, etc.) or forcibly disappearing (e.g. kidnapping and disappearances) civilian actors.
      Note that the "ViolenceAgainstCivilians" event type exclusively captures violence targeting civilians
      that does not occur concurrently with other forms of violence - such as rioting - that are coded higher
      in the ACLED event type hierarchy. To get a full list of events in the ACLED dataset where civilians
      were the main or only target of violence, users can filter on the "Civilian targeting" field.
    "ViolenceAgainstCivilians" event type has the following subtypes:
    - SexualViolence: Any event where an individual is targeted with sexual violence, including but not
      limited to rape, public stripping, and sexual torture, with the gender identities of victims recorded
      when reported.
    - Attack: An event where civilians are targeted with violence by an organized armed actor outside the
      context of other forms of violence, including severe government overreach by law enforcement.
    - AbductionOrForcedDisappearance: An event involving the abduction or forced disappearance of civilians
      without reports of further violence, including arrests by non-state groups and extrajudicial detentions
      by state forces, but excluding standard judicial arrests by state forces.
    """

    location: Location = Field(..., description="Location where the event takes place")
    targets_local_administrators: bool = Field(
        ...,
        description="Whether this violence is affecting local government officials and administrators -
      including governors, mayors, councilors, and other civil servants.",
    )
    women_targeted: List[WomenTargetedCategory] = Field(
        ...,
        description="The category of violence against women, if any. If this violence is not targeting women,
      this should be an empty list.",
    )

class SexualViolence(ViolenceAgainstCivilians):
    """
    Is a type of "ViolenceAgainstCivilians" event. This event type is used when any individual is targeted
      with sexual violence. SexualViolence is defined largely as an action that inflicts harm of a sexual
      nature. This means that it is not limited to solely penetrative rape, but also includes actions like
      public stripping, sexual torture, etc. Given the gendered nature of sexual violence, the gender
      identities of the victims - i.e. "Women", "Men", and "LGBTQ\+", or a combination thereof - are recorded
      in the "Associated Actor" field for these events when reported. Note that it is possible for sexual
      violence to occur within other event types such as "Battle" and "Riot".
    """

    fatalities: Optional[int] = Field(
        ...,
        description="Total number of fatalities, if known",
    )  # Is very very rare, only 7 events in English for 2024
    perpetrators: List[str] = Field(..., description="The attacker(s) entity or actor")
    victims: List[str] = Field(
        ...,
        description="The entity or actor(s) that is the target or victim of the SexualViolence event",
    )

class Attack(ViolenceAgainstCivilians):
    """
    Is a type of "ViolenceAgainstCivilians" event. This event type is used when civilians are targeted with
      violence by an organized armed actor outside the context of other forms of violence like ArmedClash,
      Protests, Riots, or ExplosionOrRemoteViolence. Violence by law enforcement that constitutes severe
      government overreach is also recorded as an "Attack" event.
    Attacks of a sexual nature are recorded as SexualViolence.
    If only property is attacked and not people, the event should be recorded as LootingOrPropertyDestruction
      event type.
    Excludes discovery of mass graves, which are recorded as "OtherStrategicDevelopment" events.
    """

    fatalities: Optional[int] = Field(
```

```
1782          ...,
1783          description="Total number of fatalities, if known",
         )
1784     attackers: List[str] = Field(..., description="The attacker entity or actor(s)")
1785     targeted_entities: List[str] = Field(
             ..., description="The entity or actor(s) that is the target of the attack"
1786     )
1787

1788
     class AbductionOrForcedDisappearance(ViolenceAgainstCivilians):
1789         """
         Is a type of "ViolenceAgainstCivilians" event. This event type is used when an actor engages in the
1790          abduction or forced disappearance of civilians, without reports of further violence. If fatalities or
          serious injuries are reported during the abduction or forced disappearance, the event is recorded as an
1791          "Attack" event instead. If such violence is reported in later periods during captivity, this is recorded
           as an additional "Attack" event. Note that multiple people can be abducted in a single "Abduction/
1792          forced disappearance" event.
         Arrests by non-state groups and extrajudicial detentions by state forces are considered "Abduction/forced
1793          disappearance". Arrests conducted by state forces within the standard judicial process are, however,
          considered "Arrest".
1794         """
1795
         abductor: List[str] = Field(..., description="The abductor person or group(s)")
1796     abductee: List[str] = Field(
             ...,
1797         description="People or group(s) that were abducted or disappeared. Note that multiple people can be
          abducted in a single AbductionOrForcedDisappearance event",
1798     )
1799

1800
     class StrategicDevelopment(ACLEDEvent, ABC):
1801         """
         This event type captures contextually important information regarding incidents and activities of groups
1802          that are not recorded as "Political violence" or "Demonstration" events, yet may trigger future events
          or contribute to political dynamics within and across states. The inclusion of such events is limited,
1803          as their purpose is to capture pivotal events within the broader political landscape. They typically
          include a disparate range of events, such as recruitment drives, looting, and incursions, as well as the
1804           location and date of peace talks and the arrests of high-ranking officials or large groups. While it is
           rare for fatalities to be reported as a result of such events, they can occur in certain cases - e.g.
1805          the suspicious death of a high-ranking official, the accidental detonation of a bomb resulting in the
          bomber being killed, etc.
1806     Due to their context-specific nature, "StrategicDevelopment" are not collected and recorded in the same
          cross-comparable fashion as "Political violence" and "Demonstration" events. As such, the "
1807          StrategicDevelopment" event type is primarily a tool for understanding particular contexts.
         "StrategicDevelopment" event type has the following subtypes:
1808     - Agreement: Records any agreement between different actors, such as peace talks, ceasefires, or prisoner
          exchanges.
1809     - Arrest: Used when state forces or controlling actors detain a significant individual or conduct
          politically important mass arrests.
1810     - ChangeToArmedGroup: Records significant changes in the activity or structure of armed groups, including
          creation, recruitment, movement, or absorption of forces.
1811     - DisruptedWeaponsUse: Captures instances where an explosion or remote violence event is prevented, or
          when significant weapons caches are seized.
1812     - BaseEstablished: Used when an organized armed group establishes a permanent or semi-permanent base or
          headquarters.
1813     - LootingOrPropertyDestruction: Records incidents of looting or seizing goods/property outside the context
           of other forms of violence or destruction.
1814     - NonViolentTransferOfTerritory: Used when actors acquire control of a location without engaging in
          violent interaction with another group.
1815     - OtherStrategicDevelopment: Covers significant developments that don't fall into other Strategic
          Development event types, such as coups or population displacements.
1816         """

1817     location: Location = Field(..., description="Location where the event takes place")
1818

1819
     class Agreement(StrategicDevelopment):
1820         """
         Is a type of "StrategicDevelopment" event. This event type is used to record any sort of agreement between
1821           different armed actors (such as governments and rebel groups). Examples include peace agreements/talks,
           ceasefires, evacuation deals, prisoner exchanges, negotiated territorial transfers, prisoner releases,
          surrenders, repatriations, etc.
1822     Excludes agreements between political parties, trade unions, or other non-armed actors like protestors.
         """
1823
1824     group_1: List[str] = Field(
             ..., description="Group or individual involved in the agreement"
1825     )
         group_2: List[str] = Field(
1826          ...,
             description="The other group or individual involved in the agreement",
1827     )
1828

1829
     class Arrest(StrategicDevelopment):
1830         """
         Is a type of "StrategicDevelopment" event. This event type is used when state forces or other actors
1831           exercising de facto control over a territory either detain a particularly significant individual or
```

```
1836
1837          engage in politically significant mass arrests. This excludes arrests of individuals for common crimes,
               such as theft or assault, unless the individual is a high-ranking official or the arrest is politically
1838          significant.
              """
1839
1840          detainers: List[str] = Field(
                  ..., description="The person or group(s) who detains or jails the detainee(s)"
1841          )
              detainees: List[str] = Field(
1842              ..., description="The person or group(s) being detained or jailed"
              )
1843
1844
1845      class ChangeToArmedGroup(StrategicDevelopment):
              """
1846          Is a type of "StrategicDevelopment" event. This event type is used to record significant changes in the
               activity or structure of armed groups. It can cover anything from the creation of a new rebel group or a
1847           paramilitary wing of the security forces, "voluntary" recruitment drives, movement of forces, or any
              other non-violent security measures enacted by armed actors. This event type can also be used if one
1848          armed group is absorbed into a different armed group or to track large-scale defections.
              """
1849
1850          armed_group: List[str] = Field(
                  ..., description="The name of armed group that underwent change"
1851          )
              other_actors: List[str] = Field(
1852              ...,
1853              description="Other actors or groups involved. E.g. the government that ordered a change to its army.",
              )
1854
1855
1856      class DisruptedWeaponsUse(StrategicDevelopment):
              """
1857          Is a type of "StrategicDevelopment" event. This event type is used to capture all instances in which an
               event of "ExplosionOrRemoteViolence" is prevented from occurring, or when armed actors seize significant
1858           caches of weapons. It includes the safe defusal of an explosive, the accidental detonation of
              explosives by those allegedly responsible for planting it, the interception of explosives in the air, as
1859           well as the seizure of weapons or weapons platforms such as jets, helicopters, tanks, etc. Note that in
              cases where a group other than the one that planted an explosive is attempting to render an explosive
1860          harmless and it goes off, this is recorded under the "ExplosionOrRemoteViolence" event type, as the
               explosive has harmed an actor other than the one that planted it.
1861          """
1862
1863          attackers: List[str] = Field(
                  ..., description="The entity or actor(s) responsible for the remote violence"
1864          )
              disruptors: List[str] = Field(
1865              ...,
1866              description="The entity or actor(s) disrupting the explosion or remote violence",
              )
1867          targets_local_administrators: bool = Field(
                  ...,
1868              description="Whether this violence is affecting local government officials and administrators -
              including governors, mayors, councilors, and other civil servants.",
1869          )
              women_targeted: List[WomenTargetedCategory] = Field(
1870              ...,
1871              description="The category of violence against women, if any. If this violence is not targeting women,
              this should be an empty list.",
1872          )
1873
1874
1875      class BaseEstablished(StrategicDevelopment):
              """
1876          Is a type of "StrategicDevelopment" event. This event type is used when an organized armed group
               establishes a permanent or semi-permanent base or headquarters. There are few cases where opposition
1877           groups other than rebels can also establish a headquarters or base (e.g. AMISOM forces in Somalia).
              """
1878
1879          group: List[str] = Field(
                  ..., description="Entity or group(s) establishing the base"
1880          )
1881
1882      class LootingOrPropertyDestruction(StrategicDevelopment):
              """
1883          Is a type of "StrategicDevelopment" event. This event type is used when actors engage in looting or
               seizing goods or property outside the context of other forms of violence or destruction, such as rioting
1884           or armed clashes. This excludes the seizure or destruction of weapons or weapons systems, which are
              captured under the "DisruptedWeaponsUse" event type. This can occur during raiding or after the capture
1885          of villages or other populated places by armed groups that occur without reported violence.
              """
1886
1887          perpetrators: List[str] = Field(
1888              ..., description="The group or entity that does the looting or seizure"
              )
1889          victims: List[str] = Field(
                  ..., description="The group or entity that was the target of looting or seizure"
```

```
        )
    targets_local_administrators: bool = Field(
            ...,
            description="Whether this violence is affecting local government officials and administrators -
        including governors, mayors, councilors, and other civil servants.",
    )
    women_targeted: List[WomenTargetedCategory] = Field(
            ...,
            description="The category of violence against women, if any. If this violence is not targeting women,
        this should be an empty list.",
    )

class NonViolentTransferOfTerritory(StrategicDevelopment):
    """
    Is a type of "StrategicDevelopment" event. This event type is used in situations in which rebels,
      governments, or their affiliates acquire control of a location without engaging in a violent interaction
        with another group. Rebels establishing control of a location without any resistance is an example of
        this event.
    """

    actors_taking_over: List[str] = Field(
        ..., description="The entity or actor(s) establishing control."
    )
    actors_giving_up: List[str] = Field(
        ..., description="The entity or actor(s) giving up territory, if known."
    )

class OtherStrategicDevelopment(StrategicDevelopment):
    """
    Is a type of "StrategicDevelopment" event. This event type is used to cover any significant development
      that does not fall into any of the other "StrategicDevelopment" event types. Includes the occurrence of
      a coup, the displacement of a civilian population as a result of fighting, and the discovery of mass
      graves.
    """

    group_1: List[str] = Field(
        ..., description="Group or individual involved in the StrategicDevelopment"
    )
    group_2: List[str] = Field(
            ...,
            description="The other group or individual involved in the violence, if any",
    )

class WomenTargetedCategory(str, Enum):
    CANDIDATES_FOR_OFFICE = "Women who are running in an election to hold a publicly elected government
      position"
    POLITICIANS = "Women who currently serve in an elected position in government"
    POLITICAL_PARTY_SUPPORTERS = "political party supporters"
    VOTERS = "Women who are registering to vote or are casting a ballot in an election"
    GOVERNMENT_OFFICIALS = "Women who work for the local, regional, or national government in a non-partisan
      capacity"
    ACTIVISTS_HRD_SOCIAL_LEADERS = (
        "Women who are activists/human rights defenders/social leaders"
    )
    RELATIVES_OF_TARGETED_GROUPS = "Women who are subject to violence as a result of who they are married to,
      the daughter of, related to, or are otherwise personally connected to (e.g. candidates, politicians,
      social leaders, armed actors, voters, party supporters, etc.)"
    ACCUSED_OF_WITCHCRAFT = "Women accused of witchcraft or sorcery, or other mystical or spiritual practices
      that are typically considered taboo or dangerous within some societies (excluding women who serve as
      religious leaders in religious structures that are typically not viewed as taboo or dangerous, such as
      nuns, female priests, or shamans)"
    GIRLS = "Girls who are under the age of 18; they may be specifically referred to by age or explicitly
      referred to as a child/girl"

class Location(BaseModel):
    """
    The most specific location for an event. Locations can be named populated places, geostrategic locations,
      natural locations, or neighborhoods of larger cities.
    In selected large cities with activity dispersed over many neighborhoods, locations are further specified
      to predefined subsections within a city. In such cases, City Name - District name (e.g. Mosul - Old City
      ) is recorded in "specific_location". If information about the specific neighborhood/district is not
      known, the location is recorded at the city level (e.g. Mosul).
    """

    country: str = Field(
            ...,
            description="Normalized name of a country, e.g. United States",
    )
    address: str = Field(
            ...,
            description="Full address or location description including all geographic levels upto the
        neighborhood level, including village/city, district, county, province, region, country, if available.
        Exclude street names, buildings, and other specific landmarks.",
    )
```

The languages included in LEMONADE are in Table 10.

Table 10: Mapping of language acronyms.

| Acronym | Full Name |
|---------|-----------|
| en | English |
| es | Spanish |
| ar | Arabic |
| fr | French |
| pt | Portuguese |
| ko | Korean |
| de | German |
| uk | Ukrainian |
| my | Malay |
| it | Italian |
| tr | Turkish |
| id | Indonesian |
| ru | Russian |
| fa | Persian (Farsi) |
| ne | Nepali |
| zh | Chinese |

