# OpenReview forum: "Multlingual Abstractive Event Extraction for the Real World"
_ICLR.cc/2025/Conference — Submitted to ICLR 2025_

### Official Review · Reviewer_FFB2 · 2024-11-04

**Soundness:** 2
**Presentation:** 2
**Contribution:** 2
**Rating:** 3
**Confidence:** 4

**Summary:**

This paper works on the event extraction task. Based on existing dataset ACLED, the authors construct a multilingual event extraction dataset LEMONADE. They conduct experiments on both open-source and closed-source large language models, achieving meaningful results with F1 score of 71.6%.

**Strengths:**

1. This work offers a multilingual event extraction dataset.
2. This work links the entities in the text to the corresponding entity database.
3. This work provides precise annotation of specific location information.

**Weaknesses:**

1. This work constructs an event extraction dataset but does not provide detailed data analysis, such as the distribution of event types across different languages, argument distribution, document count, and entity distribution.
2. The experimental section is insufficient and needs additional experiments to verify the effectiveness of the proposed dataset.
3. Table 4 shows an uneven distribution of event types. Is this factor considered in the experiments?
4. Table 2 presents the overall results, but how do the results vary across different event types and languages?
4. Supervised experiments based on large language models are crucial, and the authors should focus on this aspect in the methodology section.

**Questions:**

1. Need to analyze the impact for different event types and arguments.
2. The paper lacks in-depth experimental analysis, such as error analysis and case studies.
3. When constructing the training set, do the authors consider the impact of different data scales on the results?

---

### Official Review · Reviewer_kErp · 2024-11-08

**Soundness:** 3
**Presentation:** 3
**Contribution:** 2
**Rating:** 5
**Confidence:** 4

**Summary:**

The paper introduces a approach to event extraction, known as Abstractive Event Extraction (AEE), which moves beyond traditional mention-level annotations to capture a deeper understanding of events in text. They present a multilingual dataset, LEMONADE, covering 16 languages and annotated by experts for real-world socio-political event tracking. The study also introduces ZEST, a zero-shot AEE system that performs well without training data, achieving a 57.2% F1 score, and a supervised model that achieves 71.6% F1. These approaches aim to enhance event extraction in low-resource languages and complex socio-political domains

**Strengths:**

1. This paper provides a comprehensive schema for the abstractive event extraction task and, at the same time, offers a high-quality dataset for this task.
2. The authors also provide a robust framework for abstractive event extraction.

**Weaknesses:**

This paper contributes to event extraction by defining a schema for abstractive event extraction and creating a high-quality dataset. However, from my perspective, it has several weaknesses.

1. First, while the defined schema is general, it lacks specificity. For example, arguments such as "entity," "group_1," and "group_2" are extracted without a precise argument type, which may limit practicality in real-world applications. A more useful approach could involve defining argument types as an open-set extraction task, where argument types are inferred from the context rather than using general labels.

2. Second, the authors discuss some challenges in current work, such as Entity Normalization and Linking. However, not all challenges are thoroughly addressed in this paper. For instance, event coreference resolution is mentioned but not actually covered, which raises reader expectations without fully addressing the challenge.

3. Finally, as a potential regular paper at ICLR, the methodology feels relatively weak aside from the dataset contribution. The framework design is straightforward, which diminishes the overall impact of the paper's contributions.

**Questions:**

See above

---

### Official Review · Reviewer_nUV3 · 2024-11-10

**Soundness:** 1
**Presentation:** 1
**Contribution:** 1
**Rating:** 1
**Confidence:** 2

**Summary:**

The paper presented a dataset called Lemonade (Large Expert-annotated Multilingual Ontology-Normalized Abstractive Dataset of Events) for benchmarking performance on abstractive event extraction which they abbreviated as AEE. And then the paper presented a system which they call ZEST, which is a zero-shot system for AEE, which serves as a baseline for the Lemonade dataset they came up with.

**Strengths:**

Very detailed description of the Lemonade dataset developed for the AEE task and the ZEST system for providing a baseline for it.

**Weaknesses:**

Didn't really show much why Lemonade is useful and stands out among other such datasets, or show how ZEST is a performant system for AEE, as the results are not a prevailing sweep and the baseline LLaMAX is not a widely recognized system for such a task.

**Questions:**

To the ICLR community: as a community we will at some point have to address the problem of the zero entry bar for an astronomical amount of submissions, where automatically hopefully the scientific rigor and the existence of hypothesis testing are checked, before coming to the reviewers. Take this submission as an example - what hypothesis is the submission even testing?

**Details Of Ethics Concerns:**

No ethics concern.

---

### Official Review · Reviewer_HVMz · 2024-11-11

**Soundness:** 2
**Presentation:** 4
**Contribution:** 1
**Rating:** 3
**Confidence:** 4

**Summary:**

This paper claims a new design of a new task called Abstractive Event Extraction, and repurpuses an existing resource ACLED to create a dataset Lemonade for the task. Lemonade covers 16 languages, including various under-representative languages such as Burmese, Indonesian, Nepali, etc. The key difference between the new task with traditional Event Extraction task in that (1) the arguments are linked to an pre-defined domain-entity-base, rather than an text span; (2) the do not focus on extracting event trigger words. For modeling, they build a system Zest and achives 57.2% F1 with zero-shot performance.

**Strengths:**

1. Multilingual event resources are very rare, especially for under-representative languages.
2. The discovery and attempt of re-defining event extraction task is very important to the research community.

**Weaknesses:**

1. The key claim of the paper is to have a new tool for real-world EE, and thus propose the new Abstractive event extraction problem. However, the author lack the evidence to prove the how these new paradigm of task is more helpful than classical definition of event extraction. The discussion on different use cases should be included.

2. I think the claimed "novel/new" Abstractive event extraction problem is simply the previous event extraction annotation paradigm plus a "entity-linking" filtering on argument lists. For example, in Figure 1, the entities "two communties" can be and should be disambuigated by havinge an entity-linker after the traditional event extraction system. And this practice has been used in the information extraction communities for a long time. For the two event or a single event case in Figure 1, "event coreference system" are designed for this purpose. For these point of view, I do not see the claimed "Abstractive event extraction" is a new task.

3. Several of the design choice are arguable:
- Why single event for each writing is enough? If we consider diverse potential downstream applications, such as event interactions, plot understanding. Single event is far from useful.
- It is unclear to me whether intermediate annotations SHOULD be included or excluded for annotation. I think in many previous efforts, like ACE. These intermediate annotations is very important for (a.) guarantee the annotation quality (b.) evaluating step-by-step performance. In this paper, I do not see how missing these annotations influence the annotation quality and the corresponding IAA. I think this paper neglect the potential ambuiguity done by ACLED dataset.

**Questions:**

1. As mentioned in Line 248~250, the main event exluding historical events. Does it means that when applying the system in the wild, the system can only be used in extracting "novel" events?

2. In the system, an 'up-to-date' domain-specific and complete entity lists are required. How does this becomes available in real application?

---

### Meta-Review · Area_Chair_ywVA · 2024-12-21

**Metareview:**

(a) Scientific Claims and Findings
The paper defines a novel task, AEE, which emphasizes linking extracted arguments to a predefined entity ontology rather than text spans. A new dataset, LEMONADE, is introduced for this task, supporting 16 languages with a focus on low-resource settings. Two systems, ZEST (zero-shot) and a supervised baseline, are evaluated, showing moderate success. The paper aims to move the event extraction field toward a deeper, ontology-based paradigm.

(b) Strengths

- Provides a rare multilingual dataset for event extraction, covering underrepresented languages.
- Proposes a shift in the event extraction paradigm toward ontology normalization, which may have practical benefits.
- Sets initial benchmarks with the proposed zero-shot and supervised systems.

(c) Weaknesses

- The novelty of the task is questioned by reviewers, who see it as a combination of existing paradigms (entity linking + event extraction).
- The dataset lacks detailed analysis of event type distributions, entity coverage, and linguistic diversity impacts.
- Experiments are limited, with little exploration of model performance across different event types or languages.
- Claims of novelty are undermined by the absence of comparisons with existing methods or baselines widely accepted in the field.

(d) Decision: reject
While the dataset contribution is valuable, the methodological and experimental aspects of the submission are insufficiently rigorous. The task definition and evaluation lack clarity and depth, and the reviewers unanimously highlight gaps in demonstrating the task’s novelty and utility.

**Additional Comments On Reviewer Discussion:**

No rebuttal and discussion happened.

---

### Decision · Program_Chairs · 2025-01-22

Reject